

# Composition of soil fungal communities and microbial activity along an elevational gradient in Mt. Jiri, Republic of Korea

Ana Mitcov[1,*], Daegeun Ko[1,*], Kwanyoung Ko[1,2], Jaeho Kim[1], Neung-Hwan Oh[3,4], Hyun Seok Kim[5,6,7], Hyeyeong Choe[5] and Haegeun Chung[1,8]

[1] Department of Environmental Engineering, Konkuk University, Seoul, Republic of Korea
[2] Nanobio Measurement Group, Korea Research Institute of Standards and Science, Daejeon, Republic of Korea
[3] Graduate School of Environmental Studies, Seoul National University, Seoul, Republic of Korea
[4] Environmental Planning Institute, Seoul National University, Seoul, Republic of Korea
[5] Department of Agriculture, Forestry and Bioresources, Seoul National University, Seoul, Republic of Korea
[6] Interdisciplinary Program in Agricultural and Forest Meteorology, Seoul National University, Seoul, Republic of Korea
[7] Research Institute of Agriculture and Life Sciences, Seoul National University, Seoul, Republic of Korea
[8] Department of Civil and Environmental Engineering, Konkuk University, Seoul, Republic of Korea
* These authors contributed equally to this work.

Corresponding author
Haegeun Chung,
hchung@konkuk.ac.kr

## ABSTRACT

Approximately 64% of the Republic of Korea comprises mountainous areas, which as cold and high-altitude regions are gravely affected by climate change. Within the mountainous and the alpine-subalpine ecosystems, microbial communities play a pivotal role in biogeochemical cycling and partly regulate climate change through such cycles. We investigated the composition and function of microbial communities, with a focus on fungal communities, in Republic of Korea's second tallest mountain, Mt. Jiri, along a four-point-altitude gradient: 600-, 1,000-, 1,200-, and 1,400-m. Soil pH and elevation were negatively correlated, with soils becoming more acidic at higher altitude. Of the five soil enzyme activities analyzed, cellobiohydrolase, β-1,4-glucosidase, and β-1,4-xylosidase activity showed differences among the elevation levels, with lower activity at 600 m than that at 1,400 m. Soil microbial biomass correlated positively with increasing elevation and soil water content. The decrease in β-1,4-N-acetylglucosaminidase suggests a reduction in fungal biomass with increasing altitude, while factors other than elevation may influence the increase in activity of the cellobiohydrolase, β-1,4-glucosidase and β-1,4-xylosidase. Fungal alpha diversity did not exhibit an elevational trend, whereas beta diversity formed two clusters (600–1,000 m and 1,200–1,400 m). Community composition was similar among the elevations, with *Basidiomycota* being the most predominant phylum, followed by *Ascomycota*. Conversely, among the fungal communities at 1,000 m, *Ascomycota* was the most dominant, possibly due to increased pathotroph percentage. Elevational gradients induce changes in soil properties, vegetation, and climate factors such as temperature and precipitation, all of which impact soil microbial communities and altogether create a mutually reinforcing system. Hence, inspection of elevation-based microbial

communities can aid in inferring ecosystem properties, specifically those related to nutrient cycling, and can partly help assess the oncoming direct and indirect effects of climate change.

## INTRODUCTION

Approximately 64% of the Republic of Korea consists of mountainous regions, which can be further divided into subalpine areas below the tree line and alpine areas above the tree line, starting at the 1,400 m mark (*Johnson, 2004*). The overall mountainous, and the alpine and subalpine ecosystems, including those on Mt. Jiri, are sensitive to climate change, particularly as cold and high-altitude regions (*D'Alò et al., 2021*).

Microorganisms play crucial roles in subalpine ecosystems, contributing significantly to processes such as nutrient cycling, organic matter decomposition, and soil development. Specifically, soil microorganisms regulate the flow of necessary nutrients, such as phosphorus, sulfur, potassium, iron, manganese, and zinc, or contribute to nitrogen (N) cycling through transformations, such as nitrification, denitrification, and ammonification (*Mukhtar et al., 2023*). Through enzymatic activity, they decompose complex compounds, such as lignin and cellulose, contributing to the overall carbon cycle (*Prescott & Vesterdal, 2021*). Finally, they build mutualistic, commensalistic, or parasitic relationships that influence vegetation development and health (*Lv et al., 2023*). However, despite their potential role in regulating climate change, especially in colder and higher altitude ecosystems such as alpine and subalpine areas, the role of microorganisms in the climate change loop is rarely the focus of related studies (*Cavicchioli, Ripple & Timmis, 2019*).

The variations in temperature and precipitation levels induced by increases in altitude serve as an appropriate approximation of a climate gradient, where the microbiome plays an important role in biogeochemical cycling. Large-scale climatic shifts modify local vegetation and edaphic conditions, such as soil pH and soil moisture content, which further influence microbial communities (*Sun et al., 2020*). For instance, warmer temperatures increase microbial activity, leading to generally higher rates of N mineralization and nitrification. Higher N availability makes plants less inclined to form mutualistic relationships with mycorrhizal fungi, resulting in a shift in the mycobiome towards more generalized species with lower diversity (*Li et al., 2022*). The soil microbiome may be a valuable indicator of the direct and indirect effects of climate change on ecosystems, and inspection of soil microbial communities can offer insights into how changes in microbial diversity and activity patterns contribute to alterations in soil nutrient dynamics, plant-microbe interactions, and ecosystem health. Hence, as a key microbial category, this study focused on the soil-inhabiting fungal communities of Mt. Jiri, an area particularly susceptible to climate alterations as a cold and high-altitude region.

Factors influencing soil microbial communities in alpine and subalpine ecosystems have been studied for decades (*Schinner, 1982*; *Margesin et al., 2009*; *D'Alò et al., 2021*). Biotic and abiotic components, such as vegetation type, temperature, pH, moisture content, and soil type, are key determinants of the functions and diversity of microbial communities. Additionally, elevation has been reported to significantly impact microbial diversity, activity, and community composition, with relative abundances of main functional groups varying along the elevational gradient. For example, a decreasing fungi to bacteria ratio with increasing elevation was found in the Austrian Limestone Alps (900–1,900 m) (*Djukic et al., 2010*), but the opposite trend was reported in the Austrian Central Alps (*Margesin et al., 2009*). Fungal communities generally exhibited contrasting patterns, such as a decrease in diversity with increasing altitude (*Schinner, 1982*) or a hump-shaped trend with highest alpha diversity reported at mid-altitudes (*Zuo et al., 2024*). The within-community responses to elevation were likewise not uniform. In an East African Mt. Kilimanjaro study (767–4,190 m), major phyla *Ascomycota* decreased with elevation, *Glomeromycota* followed a hump-shaped curve, while *Chytridiomycota* showed a U-shaped trend (*Shen et al., 2020*). Broadly, these contradictory discoveries may be attributable to the confounding effects of regional-scale environmental factors, such as geography, rock parent material, and seasonality (*Li et al., 2022*), making it difficult to establish consensus in the literature regarding general diversity patterns or community composition. However, among different functional groups, fungal communities were shown to respond more strongly to regional-scale factors, such as mean annual temperature and precipitation, rather than local-scale factors, such as soil pH and total carbon (*Shen et al., 2020*), highlighting their importance in climate change impact studies.

To deepen our understanding of the factors shaping the soil microbial communities in alpine and subalpine ecosystems under climate change, we analyzed the soil microbial functions and fungal community composition along an elevational gradient on Mt. Jiri. Mt. Jiri is a significant natural and cultural resource in the Republic of Korea, renowned for its rich biodiversity and representative subalpine and alpine landscapes. However, it has undergone significant destruction and changes in land use, particularly in the lower regions, due to human activities including post-Korean War logging and slash-and-burn agriculture. In 1967, Mt. Jiri was designated as Republic of Korea's first National Park, and since then, substantial conservation efforts have been made to restore and protect the mountain's ecosystem (*Kim, 2024*). Thus, understanding its internal working mechanisms and the effects of climate change are of great national relevance. We aimed to address the following exploratory questions: (i) How is microbial activity on Mt. Jiri influenced by altitude? (ii) What are the dominant fungal phyla in Mt. Jiri soils and how do their relative abundances vary with elevation? (iii) How does the overall fungal diversity vary along the altitudinal gradient on Mt. Jiri? (iv) What edaphic factors predict soil fungal community composition on Mt. Jiri?

## MATERIALS AND METHODS

To establish an elevational gradient on Mt. Jiri (35°17′23.64″–35°19′26.76″N, 127°29′36.6″–127°34′11.64″E), four sampling altitudes (600, 1,000, 1,200, and 1,400 m) along the

western slope were selected (Fig. 1). Designated elevational sites belong to the permanent research station of Seoul National University. Soil samples were collected from each site in September 2021, October 2021, April 2022, and September 2022. For the 2021 samplings, we collected samples from two sites per altitude level, and for the 2022 samplings, we increased the number of sites to three per altitude level. All sites were 20 × 20 m in size and set up within a 500-m area, but the distances between them and their arrangement varied at each altitude due to the constrictive topological features present at each level. The dominant species were *Pinus densiflora* and *Acer pseudosieboldianum* at the 600-m site, *Fraxinus rhynchophylla* and *Acer pictum* Thunb. var. *mono* at the 1,000-m site, *Quercus mongolica* and *Fraxinus sieboldiana* at the 1,200-m site, and *Rhododendron schlippenbachii* and *Quercus mongolica* at the 1,400-m site. As understory vegetation, *Sasamorpha borealis* was present across all sites. Each sample was collected in two replicates and the same sampling method was used throughout. After removal of the organic layer, approximately 700 g of soils were collected between the 0 and 25 cm depth, stored in sealed bags and transported on ice to the laboratory. Soils were sieved with a 2-mm sieve and any roots, debris or residues were removed to an approximate soil weight of 500 g. Resulting soils were stored at −20 °C before starting the subsequent analyses. Due to restricted access to certain sites, we were unable to collect samples from said sites in September 2021 (1,200 m), October 2021 (1,000 and 1,400 m) and April 2022 (1,200 m). The sand, silt, and clay percentages were 41.8%, 40.4%, and 17.8% at the 600-m sites, and 36.7%, 50.2%, and 13.1% at the 1,200-m sites, respectively, with only the silt content showing a significant difference (*Pei, 2024*). The $\delta^{13}$C—soil organic carbon (SOC) was about −25‰, which is expected for forested areas with no known history of agricultural C4 plants (*Pei, 2024*). The $\Delta^{14}$C-SOC at a depth of 0–15 cm was similar between the 600- and 1,200-m sites, but at a depth of 15–30 cm it was approx. −12‰ at 600 m and −80‰ at 1,200 m, suggesting that SOC at the 1,200-m site is significantly older than that of the 600-m site (*Pei, 2024*).

## Soil analyses

Soil pH was determined using a pH meter (Mettler Toledo, Greifensee, Switzerland) after shaking 10 grams of soil in distilled water at a ratio of 1:5 (w/v) for 30 min. Water content was measured gravimetrically for 20 g of soil. Organic matter content, total N and its inorganic fractions, total carbon, and cation exchange capacity were measured at the National Instrumentation Center for Environmental Management (NICEM, Seoul, Republic of Korea).

## Soil microbial characteristics

### Soil enzyme activity

To determine microbial community metabolism, fluorometric assays were performed using methylumbelliferone (MUB)-linked substrates, as previously described in *Ko et al. (2017)*. In short, the activities of β-1,4-glucosidase (BG), cellobiohydrolase (CBH), N-acetylglucosaminidase (NAG), acid phosphatase (AP), and β-1,4-xylosidase (BX) were measured; these are extracellular enzymes involved in nutrient cycling that increase their availability or degrade cellulose, hemicellulose, and chitin in the soil (*Saiya-Cork,*
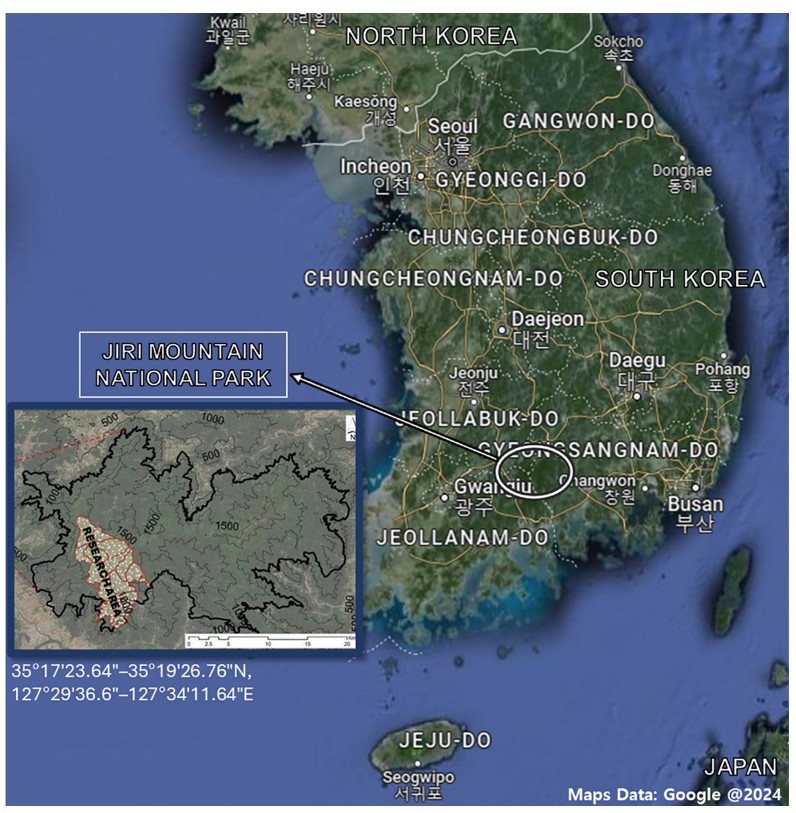

**Figure 1** **Map of the Republic of Korea and Mt. Jiri National Park with associated coordinates.** The research area was limited to four altitude levels (600-, 1,000-, 1,200- and 1,400-m) on the western slope of the mountain. Maps data: Google @2024.

*Sinsabaugh & Zak, 2002*). Two grams of soils were placed in 125 mL of sodium acetate buffer, and the slurry was transferred to a 96-well microplate that included eight analytical replicates of each enzyme assay. Plates containing all five enzymes were incubated at 24 °C for 2 h. Fluorescence was measured using a Synergy HT multi-mode microplate reader (BioTek Instruments Inc., Winooski, VT, USA), in which the excitation energy was set at 360 nm, and emission was measured at 460 nm. The enzyme activity was expressed as nmol 4-MUB g$^{-1}$ h$^{-1}$ (*Saiya-Cork, Sinsabaugh & Zak, 2002*).

### Soil microbial biomass

Soil microbial biomass carbon was measured using the chloroform fumigation-extraction method (*Vance, Brookes & Jenkinson, 1987*). Ten grams of non-fumigated and chloroform-fumigated soils were extracted using 0.5 M potassium persulfate (K$_2$S$_2$O$_8$). The C concentration was determined using a SIEVERS 900 TOC analyzer (GE Analytical Instruments, Boulder, CO, USA), and a conversion factor of 0.45 (*Wu et al., 1990*) was applied to estimate the biomass C content from the carbon concentrations recorded by the analyzer.

## Soil fungal communities

### DNA extraction and MiSeq sequencing

Soil genomic DNA was extracted using a DNeasy PowerSoil Pro Kit (Qiagen, Hilden, Germany), according to the manufacturer's instructions. The extracted soil DNA was quantified by a Nanodrop spectrophotometer (Thermo Fisher Scientific, Waltham, MA, USA) and stored at −20 °C until further use. PCR amplification was performed using the universal internal transcribed spacer (ITS) region targeting primers ITS3F (5′-GCATCG ATGAAGAACGCAGC-3′; *White et al., 1990*) and ITS4R (5′-TCCTCCGCTTATTG ATATGC-3′; *White et al., 1990*). The amplification was performed similarly to *Nam et al. (2016)*, under the following conditions: initial denaturation at 95 °C for 5 min, followed by 30 cycles of denaturation at 95 °C for 30 s, primer annealing at 55 °C for 30 s, and extension at 72 °C for 30 s, with a final elongation at 72 °C for 5 min. PCR products were confirmed using 2% agarose gel electrophoresis and visualized using a Gel Doc system (BioRad, Hercules, CA, USA). The amplified products, which were 250 bp in size, were purified using a QIAquick PCR purification kit (Qiagen, Hilden, Germany). DNA sequencing was performed by CJ Bioscience (Seoul, Republic of Korea) using the MiSeq platform (Illumina, San Diego, CA, USA) according to the manufacturer's instructions. Sequencing data are available at NCBI SRA under the project accession code PRJNA1144666.

### Fungal community analyses

The demultiplexed FASTQ files received from CJ Bioscience were inputted into the bioinformatics platform QIIME2 (version 2022.11.1) (*Bolyen et al., 2019*) and prepared for further downstream analysis.

Front and reverse reads were merged using the DADA2 plugin (*Callahan et al., 2016*) while performing quality control by trimming and truncating the sequences, denoising, and removing existing chimeras. Taxonomy was assigned to the obtained amplicon sequence variants (ASV) using the q2-feature-classifier plugin (*Bokulich et al., 2018*) employing a pre-trained Naive-Bayes classifier (*Pedregosa et al., 2011*) on the UNITE database version 9.0 (*Abarenkov et al., 2022*). The OTU tables and associated taxonomies were further imported into R version 4.3.0, where alpha and beta diversities were computed using the R package *vegan* (version 2.6-4) (*R Core Team, 2023*). We used the open annotation tool FunGuild (version 1.1) (*Nguyen et al., 2016*) to match ASVs to potentially corresponding functional guilds, with a mention of the confidence level of the match and more subdivisions, such as the trophic mode and growth morphology. The input sequences had a median value of 69,151 reads per sequence, and an average of 56.83% of the original reads was retained after quality filtering. The 24 samples had a total of 6,105 observed features, of which 2,613 were assigned to a functional guild by FunGuild.

## Statistical analyses

Statistical variance and correlation analyses were performed using SPSS statistics for Windows (version 25.0; IBM Corp., Armonk, NY, USA). Data are presented as arithmetic means with standard errors. For exploring the altitude variance, since achieving uniform
sampling across all elevation levels was challenging, the non-parametric Kruskal–Wallis test, which relies on ranks rather than means and Dunn's *post-hoc* test were employed. Dunn's test has been previously reported to reduce the effects of uneven sample size (*Elliott & Hynan, 2011*), hence was the preferred test in this study. Spearman's rank correlation was performed to test for correlations between elevation and both environmental and microbial properties. For analyses concerning variables independent of elevation level, as most violated the assumption of normality, we likewise opted for Spearman's rank correlation (*Xiao et al., 2015*). All analyses were performed at a significance level of $\alpha = 0.05$.

Multivariate analyses of fungal community were performed using the R software version 4.3.0. Principal coordinates analysis (PCoA), based on the Bray–Curtis distance, was utilized to assess the beta diversity across the four elevational levels. Additionally, Kruskal–Wallis variance testing, followed by the Wilcoxon rank–sum test as the *post-hoc* method, was employed to identify significantly different phyla and genera between the elevation levels. To account for multiple comparisons, the Benjamini–Hochberg correction was applied, as it is reported to be well-suited for noisy data such as microbial datasets (*Jiang et al., 2017*). Redundancy analysis testing (RDA) was performed to determine the environmental factors which best correlate with the dominant fungal phyla across all sites. Only variables that were relevant to the model and free from collinearity were included in the final analysis.

### Seasonal influences

Sampling was performed four times during two seasons, spring and autumn. To ensure that the observed changes in edaphic properties, microbial quantity and activity are an effect of altitude rather than seasonality, we tested for differences between seasons and found no statistically significant differences, except for soil pH at the 600-m site. DNA extraction and sequencing for fungal community analysis were performed only on soils sampled in September 2021 and 2022, and therefore were not subjected to seasonal influences.

## RESULTS

### Soil physicochemical characteristics

Soil temperature and pH in Mt. Jiri decreased significantly with increasing elevation (Spearman's correlation: $R_{temp} = -0.923$, $p < 0.001$; $R_{pH} = -0.445$, $p < 0.001$), whereas soil water content was positively correlated with elevation (Spearman's correlation: $R = 0.702$, $p < 0.001$). The soil was acidic with pH of 4.23–6.08, and soil pH was negatively correlated with soil water content (Spearman's correlation: $R = -0.689$, $p < 0.001$), with more acidic soils retaining more moisture. Significant elevation differences were identified with pH values being higher at the 600- and 1,000-m sites, and lower for the 1,200- and 1,400-m sites. The opposite trend was visible for soil water content, supporting the found inverse correlation between the two properties (Figs. 2A and 2B). Soil water content was positively correlated with CEC (Spearman's correlation: $R = 0.622$, $p = 0.031$), and TN and TC were likewise positively correlated (Spearman's correlation: $R = 0.829$, $p < 0.001$). Soil organic

matter, cation exchange capacity (CEC), total nitrogen (TN, Fig. 2C) and total carbon (TC, Fig. 2D) showed no statistically significant elevational trends.

## Microbial biomass and enzyme activity

To identify the soil microbial characteristics on Mt. Jiri, we investigated five extracellular enzymes and microbial biomass carbon. Among the activities of the five soil extracellular enzymes, cellobiohydrolase, β-1,4-glucosidase, and β-1,4-xylosidase activity exhibited differences among the elevation levels (Kruskal–Wallis: $H_{CBH}$ = 21.81, $p$ < 0.001; $H_{BG}$ = 14.75, $p$ = 0.002; $H_{BX}$ = 18.93, $p$ < 0.001) (Fig. 3B), with pairwise comparison test indicating lower activity at 600 m than that at 1,400 m ($p$ < 0.001). Soil microbial biomass was significantly lower at the 600- and 1,000-m sites than at the 1,200- and 1,400-m sites (Fig. 3A). Moreover, microbial biomass was positively correlated with elevation (Fig. 3A) and soil water content (Spearman's correlation: $R_{elevation}$ = 0.421, $p$ < 0.001; $R_{SWC}$ = 0.735, $p$ < 0.001), whereas the opposite was observed for soil pH (Spearman's correlation: $R$ = −0.590; $p$ < 0.001). To identify how microbial biomass carbon affects the trend of enzyme activity along the elevation slope, we performed the altitude variance test for enzyme activity on a gram microbial biomass carbon basis. The results of the three aforementioned enzymes remained within the same ranges. However, the activity of the β-1,4-N-acetylglucosaminidase enzyme normalized by microbial biomass carbon decreased with increasing elevation (Fig. 3B).

## Soil fungal communities

### Alpha and beta diversity

For this experiment, we clustered 517,442 quality sequences classified into 14,277 OTUs at ≥97% similarity level, distributed across all samples. The number of observed features and Shannon, Simpson, and inverse Simpson indices from the *vegan* package were used as metrics to assess fungal alpha diversity against the total feature count. An almost even distribution was observed among all the sites, with a high degree of diversity and heterogeneity. Two samples, one from the 600-m and one from the 1,000-m site, pulled the curve down, but they could be considered possible outliers existing in nature (Fig. 4A).

Additionally, the altitude variance test revealed no difference in alpha diversity among the elevation levels, and no correlation was identified with soil properties ($p$ > 0.05).

Using vegan's avgdist() algorithm at a subsampling depth of 21,900, we computed the beta diversity and observed three clusters, which may be supported by similar groupings found in the soil pH and soil moisture properties (for soils of the 600–1,000 and 1,200–1,400 m elevations) (Fig. 4B).

### Community composition

The relative abundances of the five most common phyla in Mt. Jiri were plotted, with *Basidiomycota* being the most abundant phylum accounting for approximately 45.5% of the total sequences obtained, followed by *Ascomycota* and *Mortierellomycota* at 20.6% and 14.4% of the total sequences, respectively (Fig. 5A). Figure 5B presents the nine most abundant OTUs at each altitude site, regardless of the taxonomic level, which included

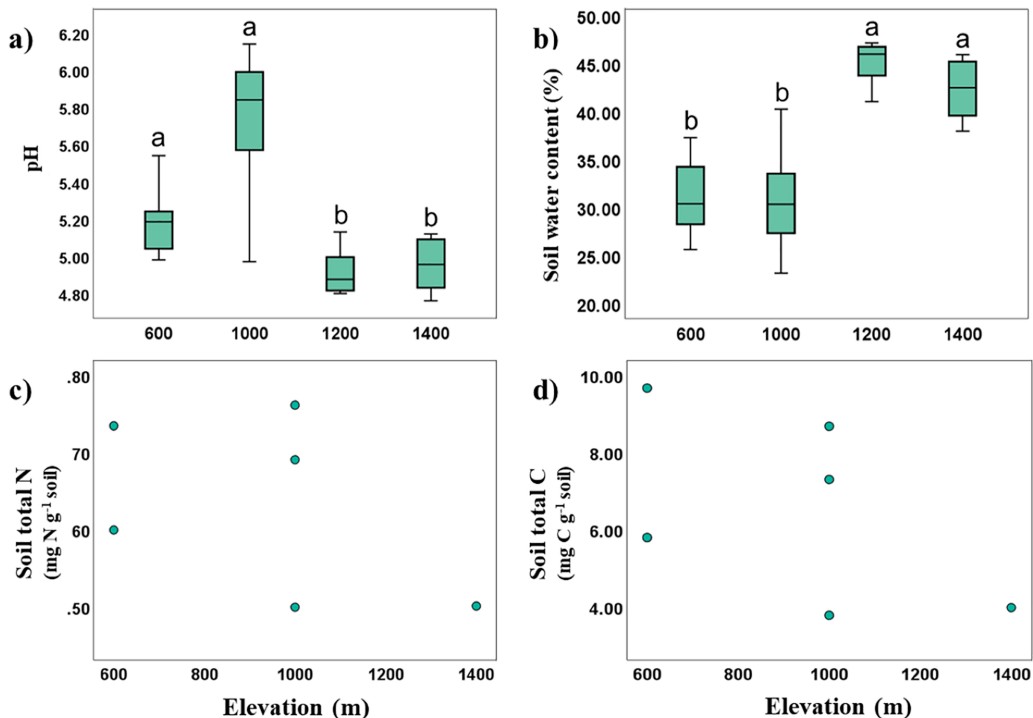

**Figure 2 Variance and correlations of soil physiochemical properties with altitude.** Elevational variance of soil pH (A), and water content (B), correlation between total N and elevation (C), and that between total C and elevation (D).

either families, such as *Mortierellaceae*, or genera, such as *Amanita*. The fungal communities in the 600-, 1,200-, and 1,400-m sites showed similar distributions of the five dominant phyla (Fig. 6), but they were also clustered with the 1,000 m-plot3 samples. This was supported by the PCoA beta diversity analysis results, which grouped the 1,000 m-plot3 samples together with those of the 600-m site (Fig. 4B). The remaining 1,000-m site samples presented relatively lower symbiotic fungi, but higher abundance of pathogenic fungi, with the latter more often belonging to the phylum *Ascomycota* (Fig. 5C). Significance testing was performed to identify statistically different phyla and initially found *Ascomycota*, *Basidiomycota* and *Olpidiomycota* to show differences. However, after applying the Benjamini-Hochberg correction, these differences were no longer statistically significant. Despite this, the elevational trend for *Ascomycota* and *Basidiomycota* can still be visualized in Fig. 6.

### Relation between main fungal phyla and environmental variables

The environmental factors, that were identified as relevant to the model and that best correlated with the dominant fungal phyla in Mt. Jiri, are graphically represented in Fig. 7. Figure 7A reflects the influence of pH, moisture, and elevation on the fungal communities present in both years of sampling, 2021 and 2022, with pH showing the strongest relation ($F = 40.26$, $p < 0.001$), followed by soil moisture ($F = 4.56$, $p = 0.039$). Additionally, we confirmed the influence of organic matter, temperature, TN, and CEC on the 2021 samples, with CEC ($F = 13.14$, $p = 0.006$), TN ($F = 8.16$, $p = 0.015$), and organic matter

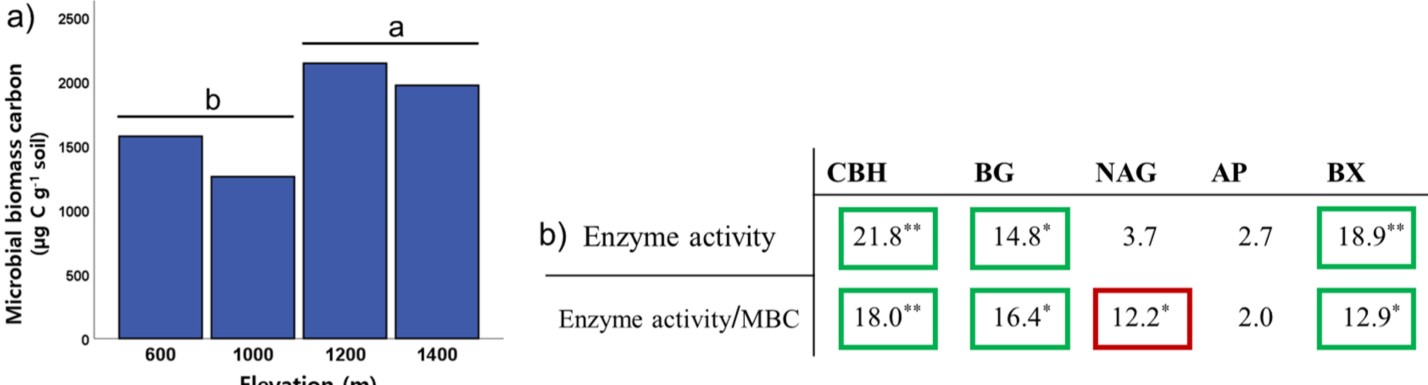

**Figure 3 Elevational trend of microbial biomass (A) and enzyme activity and enzyme activity relative to microbial biomass carbon by elevation (B).** Values in each cell represent H-scores from the Kruskal–Wallis test by elevation category with marked significance. Green square marks increase with elevation, red square marks decrease with elevation, no square marks no trend with elevation. CBH, cellobiohydrolase; BG, β-1,4-glucosidase; NAG, N-acetylglucosaminidase; AP, acid phosphatase; BX, β-1,4-xylosidase. The letters 'a' and 'b' indicate statistically different groupings at $p < 0.05$. Single and double asterisks denote statistically significant differences between elevational levels at $p < 0.05$ and $p < 0.001$, respectively.

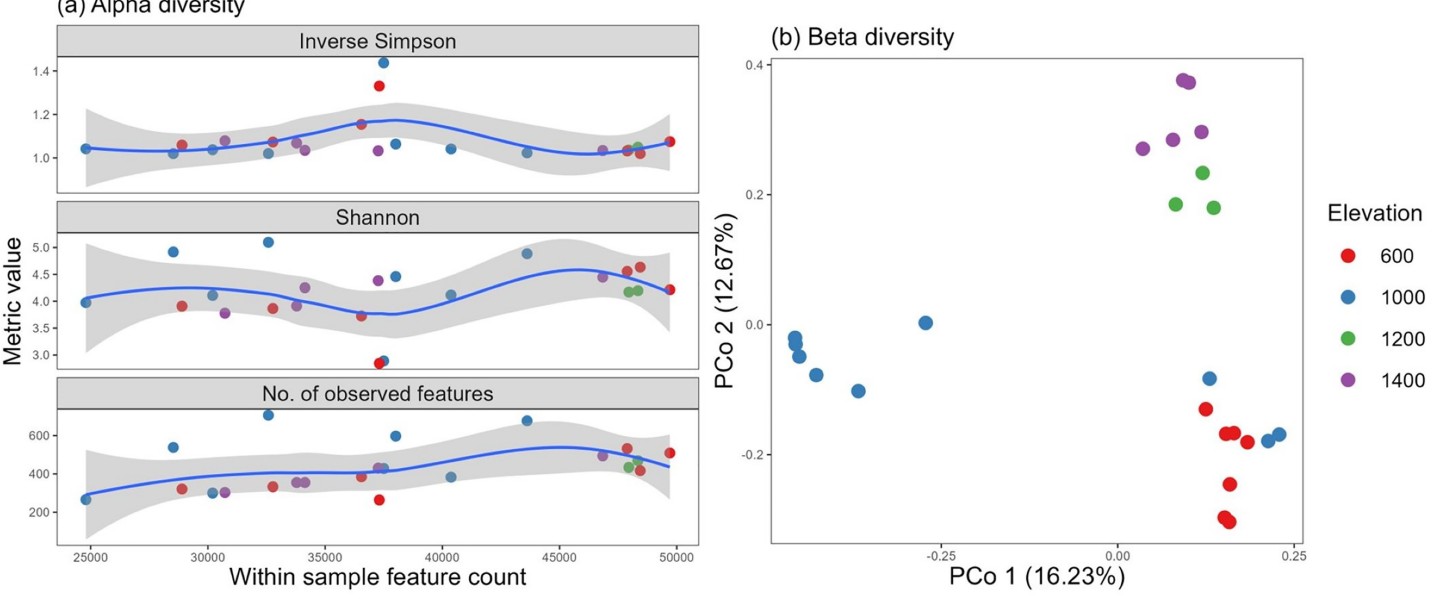

**Figure 4 Fungal alpha diversity using the inverse Simpson, Shannon, and richness metrics (A) and beta diversity (B) distributed by elevation site.**

($F$ = 5.08, $p$ = 0.052) exhibiting statistically significant relations to fungal community composition (Fig. 7B).

## DISCUSSION

In the case of Mt. Jiri, soil pH decreased with altitude, likely in response to changes in vegetation cover and increased soil moisture retention at higher elevations. An independent investigation of soil pH at 600- and 1,200-m altitudes also demonstrated a decrease of soil pH with altitude (*Pei, 2024*). In studies that have analyzed soils at higher

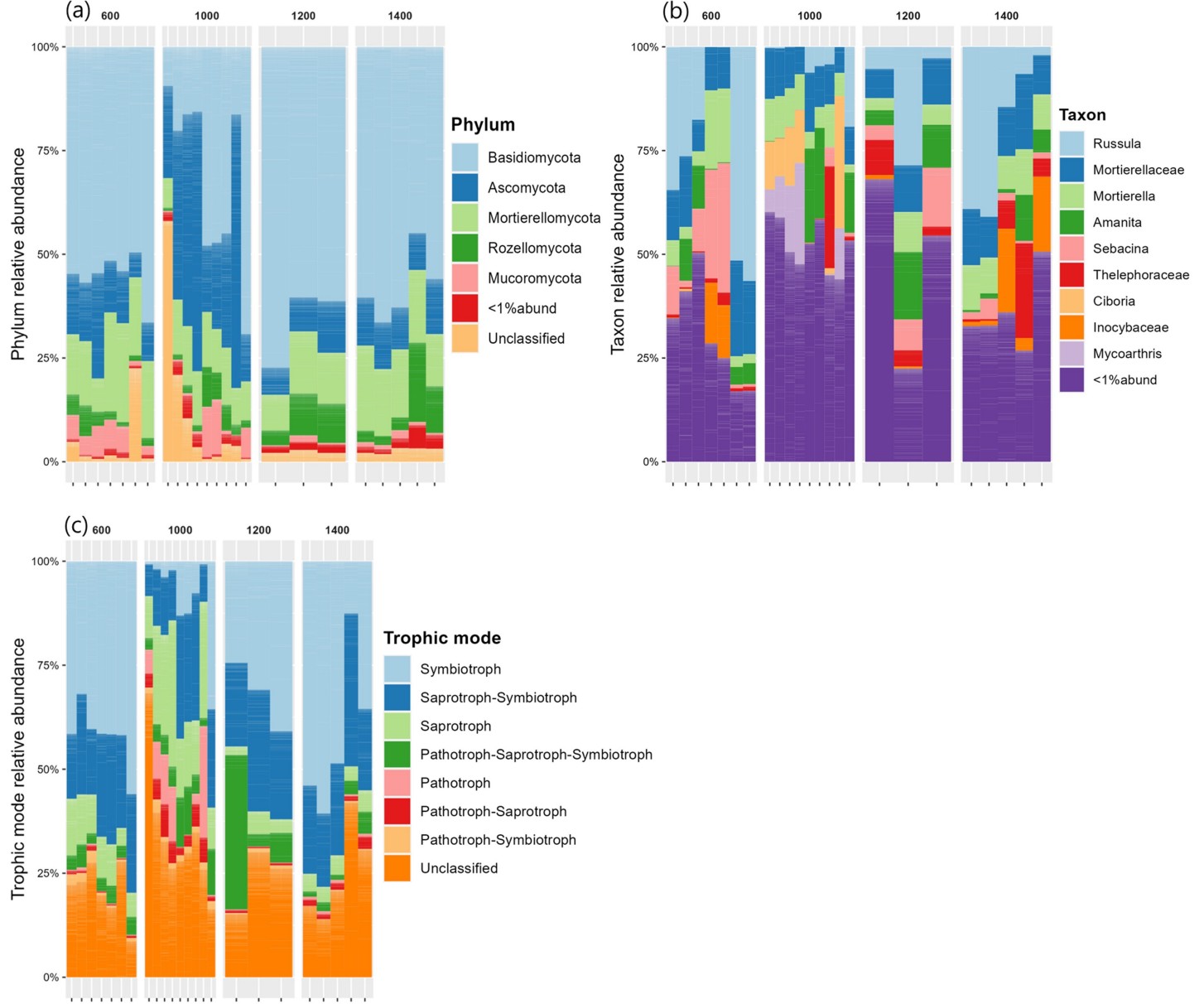

**Figure 5 Fungal phylum relative abundance (A), taxon relative abundance (B) and trophic mode relative abundance (C) distributed by elevation site.**

altitudes (3,100–5,200 m), the soil pH value decreased with an increase in elevation, possibly due to the decline in vegetation cover and increase in precipitation rates, causing leaching of basic cations (*Xu et al., 2014*; *Yuan et al., 2014*). In contrast, soil pH values increased with elevation for medium-altitude ranges (1,000–3,700 m) (*Singh et al., 2012*), whereas in other studies, no specific pattern could be observed (*Shen et al., 2013*). Although less data are available on soil moisture, the negative correlation between soil pH and soil moisture, which was also apparent in our study, was previously reported (*Yuan et al., 2014*), and moisture levels increased linearly at altitudes above the tree line

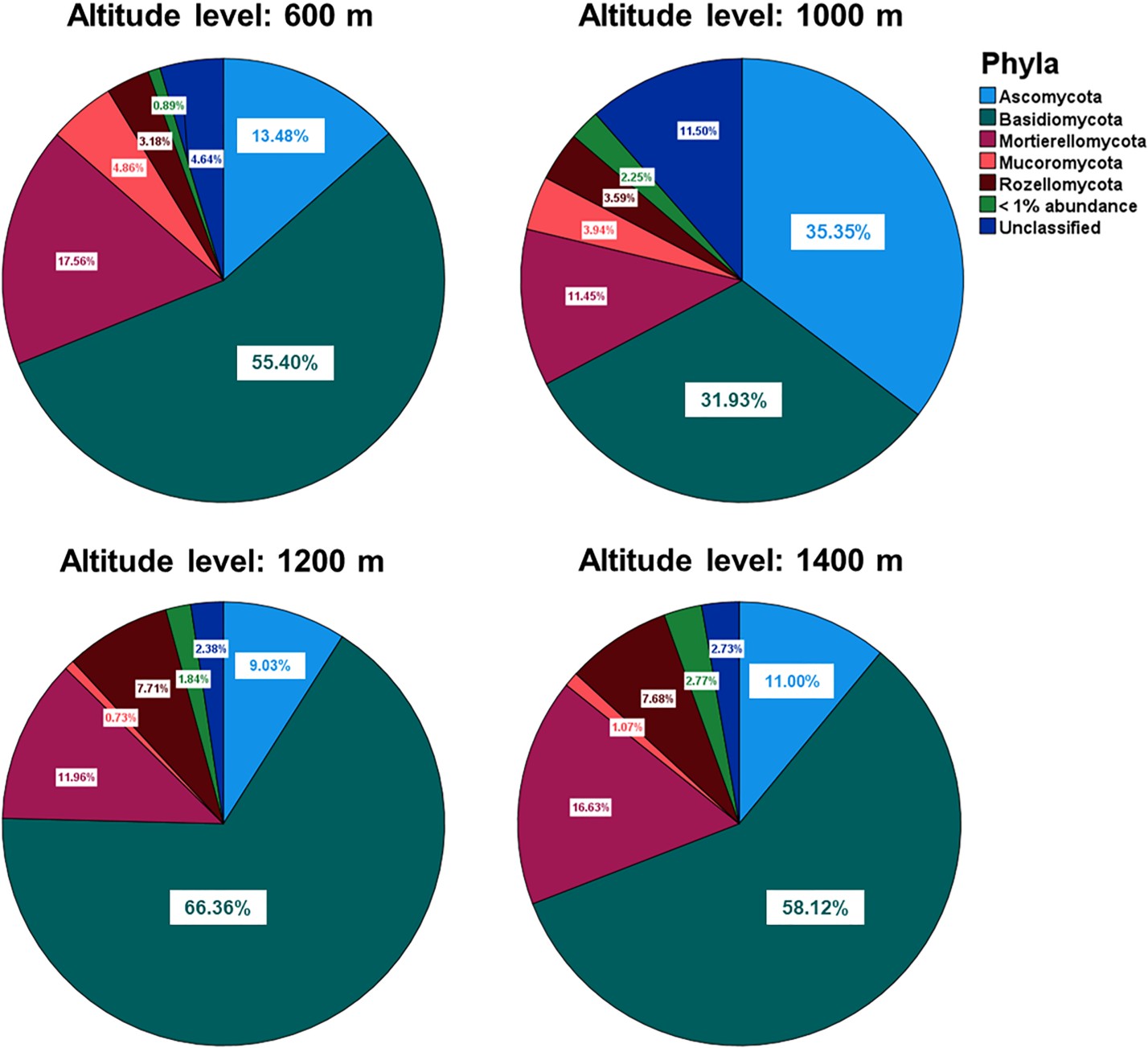

**Figure 6 Relative abundances of the five dominant phyla identified across the fungal communities of the 600-, 1,000-, 1,200- and 1,400-m sites.**
There were no statistically significant differences in the abundance of any fungal phyla across the different altitude levels.

(*Li et al., 2016*). Differences in the results reported throughout the literature may be attributed to edaphic, climatic, or region-specific differences in the study areas.

Among the five soil enzymes investigated, cellobiohydrolase, β-1,4-glucosidase, and β-1,4-xylosidase activities increased with elevation. According to *D'Alò et al. (2021)*, β-glucosidase and acidic phosphatase activities were enhanced with elevation, showing the most significant correlations with C, N, and soil microbial biomass. However, in our study,

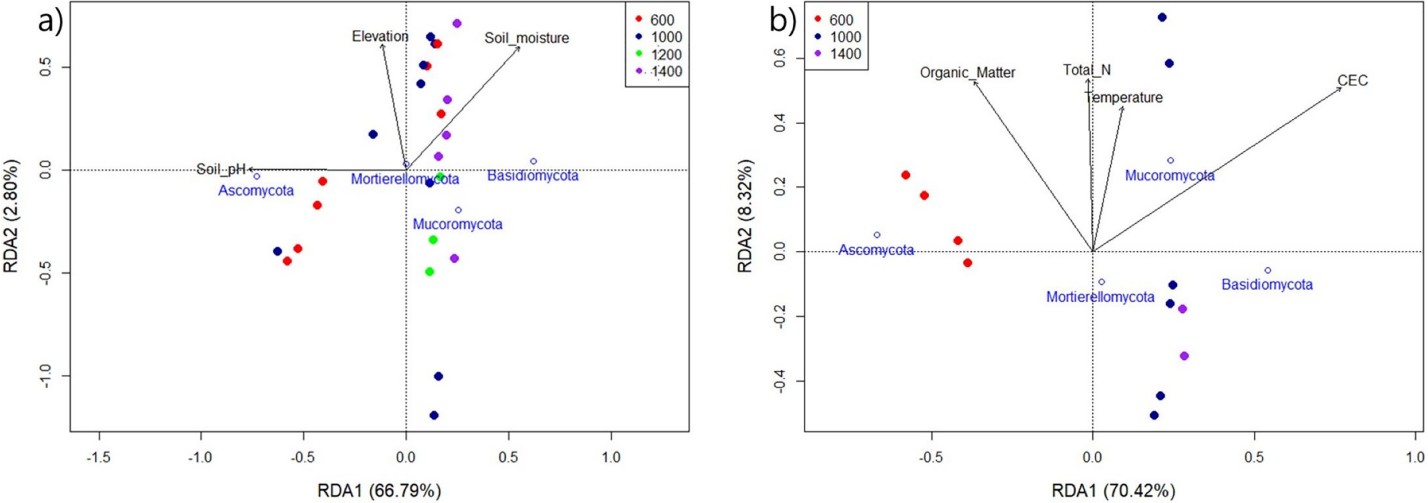

**Figure 7 Redundancy analysis of the 2021 and 2022 aggregate samples (A) and the sole 2021 samples (B) indicating the impact of environmental variables on the main fungal phyla for the four elevation sites.**

no influence of microbial biomass carbon could be observed on the activity of the three enzymes, signifying that the increase in soil enzyme activity along the elevated slope was not a result of the larger microbial biomass. Possible influencing factors include the effect of soil properties, such as soil pH and moisture, various nutrient availability, such as TN and indirectly through organic matter, or other environmental factors, such as vegetation. In the case of the β-1,4-N-acetylglucosaminidase, its activity showed no clear trend with altitude before normalizing to microbial biomass. However, when normalized, the activity decreased with increasing elevation. Since overall microbial biomass was greatest at higher altitudes and β-1,4-N-acetylglucosaminidase is often an indicator of fungal activity and biomass in the soils (*Allison, Czimczik & Treseder, 2008*), this suggests that either fungal activity lessened, or the bacteria-to-fungi biomass ratio increased in favor of bacteria at higher elevations.

In regards to fungal community composition, in the present study, the 600, 1,200 and 1,400 m soils presented similar community distribution at the phylum (Fig. 5A) and taxonomic (Fig. 5B) level with *Basidiomycota* phylum and its *Russula*, *Amanita* and *Sebacina*–genus–and *Thelephoraceae* and *Inocybaceae*–family–taxonomic subunits dominating communities at the three altitudes. *Russula* is an ectomycorrhizal symbiont that plays an important role in the global forest ecosystems (*Wang et al., 2015*) and typically thrives in neutral or acidic soils, such as is the case of the Mt. Jiri soils (*Liu et al., 2024*). Additionally, *Russula* is closely linked to tree community composition and has been found to associate with the *Pinaceae* and *Fagaceae* families (*Liu et al., 2024*), which are the primary tree families reported in Mt. Jiri (*Kim et al., 2024*). In this study, *Pinus densiflora*, belonging to the *Pinaceae* family, at the 600-m sites, and *Quercus mongolica*, belonging to the *Fagaceae* family, at the 1,200- and 1,400-m-sites, were identified as dominant species, hence explaining the prevalence of the genus at the three altitudes. In both the Northern Limestone and the Central Austrian Alps, the class *Agaricomycetes* decreased with elevation (*Bhople et al., 2022*). In this study, the trend was reflected by the mycorrhizal

symbiont genus *Sebacina*, which decreased gradually at the 600-, 1,200-, and 1,400-m sites, unlike the other genus belonging to the class *Agaricomycetes*, *Russula*. The higher abundance of the ectomycorrhiza-rich *Thelephoraceae* and *Amanitaceae* families, likewise, contributed to the dominance of phylum *Basidiomycota*. In the Northern Limestone and the Central Austrian Alps, symbiotrophs were most abundant at lower elevation sites (900, and 1,300 m, respectively) and were gradually replaced by saprotrophic fungi at middle and high elevations (1,300–1,900, and 1,600–2,100 m, respectively) (*Bhople et al., 2022*). The same elevational trend could not be observed in our research, but it was the case of the 1,000-m site, where *Ascomycota* phylum, with the saprotrophic genus *Ciboria* (*Lumbsch & Huhndorf, 2007*) and *Mycoarthris* (*Marvanová et al., 2002*) not present at other altitudes, dominated. Moreover, this study marks one of the rare records of *Mycoarthris* in the Republic of Korea, previously being recorded in fresh waters (*Lim, Nguyen & Lee, 2021*). Other abundant saprotrophic taxon was genus *Mortierella*, that remained constant at all altitudes of the gradient. Additionally, the beta diversity analysis revealed clustering between the 1,000 m-plot3 and the 600-m-sites, with dominant taxa being shared between them. While the exact cause of this clustering is unclear, it is worth noting that unlike 1,000 m-plot1 and -plot2, which are situated close to one another, plot3 is located in a more remote area at a slightly lower altitude.

Generally, environmental factors, such as pH, soil moisture, soil organic carbon or various nutrients have been shown to shape fungal communities in elevational gradients (*Zhou et al., 2021*). For example, in the Eastern Andes, Peru, the fungal alpha diversity in the mineral horizon decreased linearly with elevation, with mean annual temperature as the deterministic factor, whereas fungal alpha diversity in the organic horizon followed a concave shape with the lowest point in mid-altitude (*Nottingham et al., 2018*). In Norikura Mountain, Japan, overall diversity showed a dip along the elevation gradient, with the lowest value in the middle around 1,700 m, with the two most influential factors being the elevation gradient and mean annual temperature. However, the main phyla present— *Ascomycota*, *Basidiomycota*, *Chytridiomycota*, and *Zygomycota*—showed a linear increase in abundance with higher elevation (*Ogwu et al., 2019*). Additionally, the fungal co-occurrence network, which depicts species as nodes and relationships for matter, energy, or information exchange as links, indicated towards decreased connectivity, with fewer links observed with increasing altitude. It also showed fewer keystone taxa, marked by fewer network nodes, compared with those at lower elevations (*Yang et al., 2021*). This indicates a less compact fungal network structure at higher altitudes, potentially because of decreased vegetation diversity and enhanced environmental stress, which manifests through soil physical properties. In this study, we observed similar components, such as soil moisture, organic matter, total N, temperature, and CEC, to be influential in Mt. Jiri (Fig. 7). However, we discovered pH to be the leading driver of community changes, finding supported by other research (*Liu et al., 2018*; *Bhople et al., 2022*). Additionally, pH exhibited a close relation with phylum *Ascomycota* (Fig. 7A) and the two main phyla, *Ascomycota* and *Basidiomycota*, had opposing responses to the environmental factors, which was reported in previous studies (*Aqeel et al., 2024*).

In a study that investigated fungal community differences based on the health of the Korean fir tree species on Mt. Halla in the Republic of Korea, the same three main phyla present in our research were identified; however, *Ascomycota* had the highest percentage, followed by *Basidiomycota*, and *Mortierellomycota* (*Jeong et al., 2023*). Similarly, a higher abundance of *Ascomycota* was associated with the increased presence of pathogenic fungi in bulk soil and the rhizosphere of dead Korean fir trees, such as is the case of the 1,000-m soils in our study (Fig. 5C).

Overall, in Mt. Jiri an increasing trend in fungal activity with elevation was observed and *Basidiomycota, Ascomycota* and *Mortierellomycota* were identified as the predominant phyla. However, their relative abundance did not show any statistically significant elevational trend and neither did the alpha diversity at the four altitude levels. Both local factors such as soil pH, total N, organic matter content and CEC, as well as regionally influenced factors, such as soil water content and temperature were found to influence soil fungal communities. Hence, our study adds to the understanding that the diversity, structure, and driving mechanisms of fungal communities in alpine and subalpine ecosystems may be influenced by a vast number of contributing factors, leading to no universal pattern along the elevation slope.

## CONCLUSIONS

In this study, we investigated the elevation gradient from 600- to 1,400-m on the second tallest mountain in the Republic of Korea, Mt. Jiri, and analyzed the soil properties and microbial community trends with elevation. Elevation was negatively correlated with soil pH, with soils becoming more acidic at higher altitudes. In addition, we confirmed the negative correlation between soil pH and soil moisture, the latter of which increased with elevation. These trends may be attributed to meteorological conditions, such as higher precipitation rates at higher altitudes leading to increased moisture, or the changes in vegetation cover. Microbial biomass also increased with elevation, and cellobiohydrolase, β-1,4-glucosidase, and β-1,4-xylosidase showed increased activity at higher elevations. However, no correlation was found between microbial biomass and enzyme activities, signifying that the increase in microbial biomass did not correspond to higher soil enzyme activity. Instead, it can be inferred as a byproduct of the effects of pH, soil moisture, CEC and TN—environmental factors designated through RDA analysis as impacting community composition. Fungal alpha diversity showed no elevational trend, but did indicate a stable, rich fungal community throughout Mt. Jiri, which had a different community composition with diversification observed at mid-altitudes (two clusters at 600–1,000 and 1,200–14,000 m elevations). Long-term monitoring and further comprehensive analyses of vegetation and soil biogeochemical properties are recommended to reveal the main factors controlling soil microbial community composition in the subalpine areas of Mt. Jiri.

## ACKNOWLEDGEMENTS

We thank Daniel Ha, Minji Jin, Jaeyeon Kwon, and Solin Lee for their help with sampling during fieldwork and Chanoh Park for his help collecting and providing vegetation data.

### Funding

This research was supported by the National Research Foundation of Korea (NRF) grant funded by the Korean government (Ministry of Science and ICT) (No. NRF-2021R1A4A1025553). The funders had no role in study design, data collection and analysis, decision to publish, or preparation of the manuscript.

### Grant Disclosures

The following grant information was disclosed by the authors:
National Research Foundation of Korea (NRF) grant funded by the Korean Government (Ministry of Science and ICT): NRF-2021R1A4A1025553.

### Competing Interests

The authors declare that they have no competing interests.

### Author Contributions

- Ana Mitcov performed the experiments, analyzed the data, prepared figures and/or tables, authored or reviewed drafts of the article, and approved the final draft.
- Daegeun Ko conceived and designed the experiments, performed the experiments, analyzed the data, authored or reviewed drafts of the article, and approved the final draft.
- Kwanyoung Ko performed the experiments, authored or reviewed drafts of the article, and approved the final draft.
- Jaeho Kim performed the experiments, authored or reviewed drafts of the article, and approved the final draft.
- Neung-Hwan Oh conceived and designed the experiments, authored or reviewed drafts of the article, and approved the final draft.
- Hyun Seok Kim conceived and designed the experiments, authored or reviewed drafts of the article, and approved the final draft.
- Hyeyeong Choe conceived and designed the experiments, authored or reviewed drafts of the article, and approved the final draft.
- Haegeun Chung conceived and designed the experiments, authored or reviewed drafts of the article, and approved the final draft.

### DNA Deposition

The following information was supplied regarding the deposition of DNA sequences:
The sequencing data are available at NCBI SRA: PRJNA1144666.

### Data Availability

The data is available in the Supplemental Files.
The sequencing data are available at NCBI SRA: PRJNA1144666.

## Supplemental Information

Supplemental information for this article can be found online at http://dx.doi.org/10.7717/peerj.18762#supplemental-information.

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
