# Peer review of "Composition of soil fungal communities and microbial activity along an elevational gradient in Mt. Jiri, Republic of Korea"

_PeerJ, doi:10.7717/peerj.18762_

## Round 0.1 · original submission · Major Revisions

Your manuscript has now been reviewed by three external reviewers. The reviewers think that your work is of potential interest; however, they also raised several concerns about your study. In particular, multiple reviewers have concerns about (1) the suitability of the title, and (2) the lack of description of the study sites, and (3) the content/structure of your manuscript. Please address these outstanding issues in your revised manuscript.

·

Basic reporting

There are still many defects and deficiencies in the current manuscript, and the overall data analysis and content writing need to be greatly improved before publication.

Title: The title cannot accurately reflect the main content of this study and needs to be rewritten.

Abstract: the whole abstract should be re organized and rewritten. At the beginning of the abstract, it is enough to write about the key scientific issues, and at the end, “In addition to our study, we recommend long-term monitoring and further comprehensive analyses of vegetation and soil biogeochemical properties to reveal the main factors controlling soil microbial community composition in the subalpine areas of Mt. Jiri.” is not necessary, should be deleted.

Put “2.4. Statistical analyses” after “2.5. Fungal community analysis”.

Experimental design

The experimental design lacks the standard design for the selection of sample sites at different altitude gradients, such as vegetation type, soil type, topographic factors, etc., and the collection of habitat environmental information at sampling sites.

Validity of the findings

Data analysis and mining were insufficient, and many important results were not presented, such as the variance test of environmental indicators at different altitudes, the composition and structure of soil fungal communities at different altitudes, and the analysis of differences between communities.

Additional comments

The manuscript needs to be substantially revised and improved before it can be submitted again.

Reviewer 2 ·

Basic reporting

The paper is well-structured, with interesting findings on soil fungi in South Korea. However, the title and introduction should reflect the specific focus on fungi more clearly. The term 'soil microbiome' encompasses a wide range of living microbes related to soil. The introduction provides important background on the microbiome and its relationship with plants, physicochemical properties, climate, and biogeographical trends, offering examples of bacteria and fungi that give insights into the study’s target organisms. However, it is only in the last paragraph that the fungi are identified as the main focus of the study. I suggest revising the introduction to emphasize the importance of fungi from the outset.

Experimental design

The experimental design was appropriate to the paper's main goal, and the research questions were clear and well-related to the methods used.

Validity of the findings

The findings were appropriate for the ecosystems, vegetation, and soil physicochemical properties. Conclusions were drawn based on the results and discussion and provide a good approximation for understanding fungal ecology in South Korea.

Annotated reviews are not available for download in order to protect the identity of reviewers who chose to remain anonymous.

·

Basic reporting

This article is written and cited well and I believe the topic is of relevance to a wide readership. However, there are some issues with reporting of results and methods, that although are major, I believe can be easily addressed.

I am not familiar with this specific journal formatting requirements so excuse me if these are wrong. First, I do not see any benefit of reporting statistical tests in the abstract. The addition of P values and R2 values are not necessary. It is enough to just say 'was significant'/'was not significant' etc.

Second, the reporting of methods in the results is inappropriate. For example, at lines 203 - 204, 212 - 214, 220 - 223, and 237 - 240. There might be other instances I've missed. The description of your statistical tests should be in the methods and results should have purely the outcome of those tests. Please also provide a justification (with references where appropriate) for those analyses.

Lastly, minor point but I would like to see more context for the study sites. I, and probably many readers, would not be familiar with this particular national park. Can you comment on its history at all? For example, if there is an agricultural legacy. This would have a major impact on soil microbiomes and should be considered.

Experimental design

My major comment for the experimental design is whether or not 'time' was considered in your models? Microbiomes can vary significantly spatiotemporally. It looks like you took 4 different samples across a years' timespan. What was your justification for combining those samples? Would results be different if you added 'time' as a variable to your models? Please provide comment on why you did not look at time as a response variable.

Some parts of the methods were a bit confusing too. For example, at line 99 - 100. When you say there were two 'sites' per altitude, how far apart were sites? were they same at each altitude?

You could also provide more detail on the methods at line 100 -101. Here you talk about soil collection, but you do not detail how much soil was taken.

Validity of the findings

I think your findings are interesting, but I question the structure of your paper and how this relates to your findings. For example, throughout the abstract and introduction you rightly link elevational gradient studies to climate change consequences, e.g., using elevational gradients to understand how climate change may affect communities. That was a great set up for the discussion. However, I can't really see anywhere in the discussion how you link your results back to that literature. Your discussion is mostly just describing your results without diving too much into the literature and looking at 'the bigger picture'. I would prefer if your discussion took a wider look at the importance of your results. This should be sprinkled throughout, but would mainly sit in the conclusions.

Also, for your Figure 4 there are three sites at 1000 m that are clustered with your 600 m communities. This seems like a very interesting result and they stand out to me. What was it about those sites that made them not like the other 1000 m sites? Please discuss.

Additional comments

Line 93 - 95: Not sure why you use 'simulated' here, considering that you actually used an elevational gradient. It was not simulated.

---

## Round 0.2 · Minor Revisions

A reviewer still has concern about one of their major comments.

·

Basic reporting

Improved

Experimental design

Improved

Validity of the findings

Improved

Additional comments

The detail added to the manuscript upon revision has alleviated many of my original concerns. The authors have done well to improve the quality of the manuscript, in particular adding a much finer level of detail.

However, I still have a concern with the analyses that I believe was not addressed. Particularly my comment #3.8 surrounding why you chose to not include 'time' as a variable in the model. The graphical evidence you have provided hasn't convinced me 'time' would not be important for your models. The authors have shown pH, water content, and MBC over time to show how microbial communities likely would not have changed much over the 4 sampling seasons. These are just proxies. You have the fungi data there (alpha/beta diversity). What do these look like when you observe them over time? Is there much change? Is there an interaction with elevation?

Again, I'm not suggesting that you definitely need to re-run the analysis. But, I think you need further justification for combining your samples, considering microbes vary highly spatiotemporally. If that's why you wanted to combine them, then provide justification in the text.

---

## Round 0.3 · accepted · Accept

The authors have addressed all of the reviewers' comments and this manuscript is now ready for publication.